# Spin-decoupling of vertical cavity surface-emitting lasers with complete phase modulation using on-chip integrated Jones matrix metasurfaces

Pei-Nan Ni[1], Pan Fu[2], Pei-Pei Chen [3] ✉, Chen Xu [2], Yi-Yang Xie [2] ✉ & Patrice Genevet [1] ✉

Polarization response of artificially structured nano-antennas can be exploited to design innovative optical components, also dubbed "vectorial meta-surfaces", for the modulation of phase, amplitude, and polarization with subwavelength spatial resolution. Recent efforts in conceiving Jones matrix formalism led to the advancement of vectorial metasurfaces to independently manipulate any arbitrary phase function of orthogonal polarization states. Here, we are taking advantages of this formalism to design and experimentally validate the performance of CMOS compatible Jones matrix metasurfaces monolithically integrated with standard VCSELs for on-chip spin-decoupling and phase shaping. Our approach enables accessing the optical spin states of VCSELs in an ultra-compact way with previously unattainable phase controllability. By exploiting spin states as a new degree of freedom for laser wavefront engineering, our platform is capable of operating and reading-out the spin-momentum of lasers associated with injected spin carriers, which would potentially play a pivotal role for the development of emerging spin-optoelectronic devices.

The exploitation of electron spin degree of freedom as information carrier in addition to the charge degree of freedom has grown into a vibrant research field, known as "spintronics", which promises novel multifunctional electronic devices with non-volatility, higher operation speed and very low power consumption, etc[1,2]. Spin-optoelectronics that explicitly utilizes the spin of an electron-hole pair inside direct band-gap semiconductor opens a novel platform to transfer the quantum information stored in the spin states of electrons to the spin angular momentum of the emitted photons via the optical selection rule. The emergence of spin-optoelectronics establishes close connections between the spintronics and optoelectronics with

appealing potential for the implementation of innovative optoelectronic systems. In this context, comprehensive studies of spin injection, transport, and coherence of electrons has fueled the rapid growth of spin-polarized light source, including spin light-emitting diodes (spin-LEDs) and spin-controlled semiconductor laser (spin-lasers)[3–5]. Crucially, the exchange of quantized information from matter to the circular polarization states of photons, and the design of spin-selective optoelectronic devices will benefit from spin-dependent photonic interfaces[6,7]. During the past few years, remarkable achievements have been made in terms of manipulating and/or reading-out the electron spin using optical interfaces, including photonic waveguides[8,9],

[1]Université Côte d'Azur, CNRS, Centre de Recherche sur l'Hétéro-Epitaxie et ses Applications (CRHEA), Valbonne, France. [2]Key Laboratory of Opto-electronics Technology, Ministry of Education, Beijing University of Technology, Beijing, China. [3]Nanofabrication Laboratory, CAS Key Laboratory of Nanophotonic Materials and Devices, National Center for Nanoscience and Technology, Beijing, China. ✉e-mail: chenpp@nanoctr.cn; xieyiyang@bjut.edu.cn; patrice.genevet@crhea.cnrs.fr

plasmonic interfaces[10], and nonlinear metasurfaces[11]. However, it is challenging to combine the waveguide approach with the current spin-polarized sources due to their poor mutual integrability. On the other hand, despite that the plasmonic nanostructures have the advantage of ease of integration, their high dissipative losses present a major hurdle for the practical use.

Dielectric metasurfaces, which are interfaces composed of two-dimensional (2D) array of artificially patterned lossless optical resonators, provide flexibility and selectivity in spin-polarized wavefront engineering. In particular, this new type of planar optics enables highly efficient control of the amplitude, phase, and polarization states of light with exceptional subwavelength spatial resolution, adding a new paradigm into optical design for a large variety of ultracompact optical components, including metalens with large numerical aperture[12], waveplates[13], polarizers[14], and holograms[15,16], etc. Recent advances in metasurface polarization optics, also referred as Jones matrix metasurfaces, demonstrate remarkable controllability over the spatial distribution of phase, amplitude and polarization states, including one-only circular polarization modulation near a singularity[17–19]. Different from the previous design of metasurface, Jones matrix metasurface can independently tailor the phase profiles on any pair of orthogonal polarizations by combining the geometric phase, also called Pancharatnam-Berry (PB) phase, with the propagation, resonant or singular phase to break the phase conjugation limitation of using geometric phase alone[20,21], and thus allow for the implementation of versatile multiplexing metasurfaces with dual functionalities[22–24]. Nevertheless, to date, metasurfaces have been mainly used as stan-dalone devices for free-space applications, for which an external light source and deliberate irradiation alignments are needed. The notice-able advantages of metasurfaces for integrations with photonic devi-ces that root in their unique characteristics, such as planar configuration, ultra-compactness, compatibility with complementary metal oxide semiconductor (CMOS) process, and lightweight, have been highly appreciated and are triggering a new surge of integrated-metadevices in numerous platforms, including optical fibers[25,26], waveguides[27,28], semiconductor light sources[29,30], and CMOS chips[31], etc. Notably, the territories of on-chip optoelectronic integrations of metasurfaces can be considerably expanded to implement wafer-level sorting and wavefront shaping of spin-decoupled semiconductor lasers by further exploring the design methodology of Jones matrix metasurfaces.

With respect to edge-emitting lasers, vertical cavity surface-emitting lasers (VCSELs) exhibit several unique characteristics, such as small threshold current, circular beam profile, high modulation speed, and large-scale 2D array[6,32], making them highly preferred for spin-optoelectronic applications. In addition, the short cavity length of VCSELs (typically less than 100 nm) favors the escape of spin-polarized photons with much less reabsorption, facilitating the access to the spin-polarized photons using optical polarization components. How-ever, conventional polarization optics is usually bulky and heavy, and requires a precise alignment process for integration, which forfeits the advantage of compactness for which VCSELs are renowned. By con-trast, polarization metasurface optics not only have the advantages of ultra-compactness, lightweight and relatively high efficiency, they also provide a powerful tool to simultaneously control both the polariza-tion states and phase profiles, in line with the trend toward miniatur-ized optical systems. Despite that the spin-decoupled polarization metasurfaces have been widely engaged into free-space applications, their integration with VCSELs has not been explored to date.

Here, we propose to integrate Jones matrix metasurfaces for the purpose of optical spin-decoupling of VCSEL emissions. Beside the capability to on-chip decouple the circularly polarized (CP) states of the laser beam, we aim as well to arbitrarily modulate the phase profiles of each spin component independently, including dual-channel holographic images projection, directional generation of

multi-collimated beams with polarization dependence. Based on the spin-decoupled CP beams as the basis, we further demonstrate the feasibility to explicitly manipulate the polarization states of VCSELs. As a proof of concept, vector beams with radial and azimuthal polarizations were generated, respectively. Fully combining the unique properties of metasurface polarization optics, such as the compactness, multi-functionalities, and polarization controllability, this work unlocks a generalized scheme for on-chip beam structuring of semiconductor lasers with complete polarization degree of free-dom. It also proves that vectorial metasurfaces could play a sig-nificant role to read-out and manipulate the spin-polarized photons at an ultracompact device level, thus highlighting their important potential for spin-optoelectronic applications.

## Results
### Design and integration principles

The transmission coefficients of a birefringent meta-atom can be conveniently described using Jones matrix as[20]:

$$J(\theta) = R(-\theta)\begin{bmatrix} t_u & 0 \\ 0 & t_v \end{bmatrix} R(\theta) \tag{1}$$

where $t_u = T_u e^{i\varphi_u}$, and $t_v = T_v e^{i\varphi_v}$ are the complex transmission coefficients of the meta-atom along its fast and slow axes, respectively, $R(\theta) = \begin{bmatrix} \cos\theta & \sin\theta \\ -\sin\theta & \cos\theta \end{bmatrix}$ is a rotation matrix, $\theta$ is the rotation angle relative to the reference coordinate, as depicted in Fig. 1a. Upon the incident CP light $E_{in}$ with different spin states $|\sigma^+\rangle$ and $|\sigma^-\rangle$, where $|\sigma^+\rangle$ represents the right-circularly polarized (RCP) light and $|\sigma^-\rangle$ the left-circularly polarized (LCP) light, the transmitted light $E_{out}$ can be expressed as:

$$E_{out} = J(\theta) \cdot E_{in} = J(\theta) \cdot |\sigma^\pm\rangle = \frac{t_u + t_v}{2}|\sigma^\pm\rangle + \frac{t_u - t_v}{2}e^{\mp i2\theta}|\sigma^\mp\rangle \tag{2}$$

where the first term represents the co-polarized component with the same spin state as the incident light, and the second term stands for the cross-polarized component with the opposite spin state imposed by an additional PB phase. Equation 2 clearly shows that the full 2π phase coverage can be easily achieved for the cross-polarized light by rotating the anisotropic meta-atom from 0° to 180°. Capitalizing on the PB phase mechanism, metasurfaces with various functionalities, such as achromatic metalens[12,33], holograms[34,35], and polarization tunable devices[36,37], etc have been widely demonstrated using an array of anisotropic meta-antennas that have the same geometry but spatially varying rotation angles. However, there are some restrictions if the PB phase is used alone: Firstly, the same structure will impose equal and opposite phase profiles on incident beams of different spin states. Such conjugate phase function limits their applications. For example, Wen et al. demonstrated the deflection of opposite CPs components of VCSEL by equal and opposite angles through on-facet integration of PB metasurface[38]. This approach is not able to simultaneously collimate both transmitted beams since a collimating geometric phase works as converging lens for one spin state only, leaving the other operating as a diverging lens. Secondly, according to Eq. 2, only the cross-polarized light picks up the PB phase. Therefore, to achieve phase modulation with high efficiency, each meta-atom of the PB phase metasurface must work as an ideal half-wave plate, with the conditions: $|\varphi_u - \varphi_v| = \pi$, and $|T_u| = |T_v|$, otherwise a significant amount of light will stay at its original spin state, i.e., the co-polarization, of which the phase remains unmodulated.

Consequently, to independently tailor the phase of both spin states of the VCSEL with unity conversion efficiency, the Jones matrix

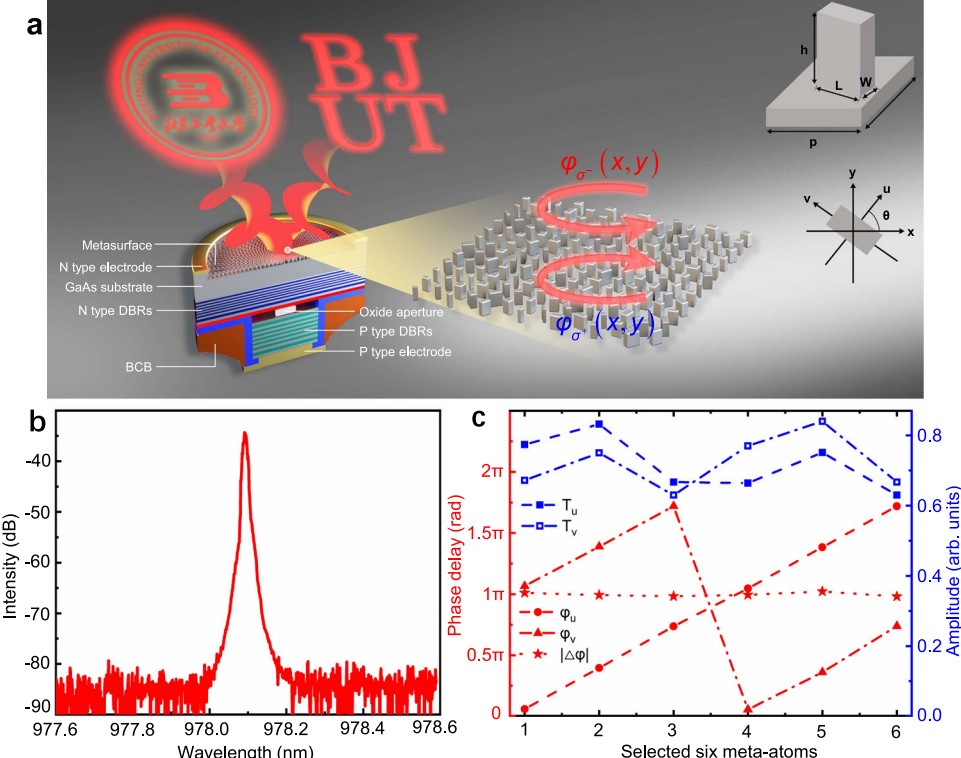

**Fig. 1 | Conceptual design and integration principles. a** Schematic illustration of a spin-decoupled VCSEL via on-chip integration of a Jones matrix metasurface composed of nanofins, enabling imposing independent and arbitrary phase profiles to the orthogonal CPs beams. In this example, spin states were encoded to display two different holographic images in the far field. **b** Laser peak with small full width at half maximum (<0.02 nm) and high side-mode suppression ratio (>40 dB) were observed from the emission spectrum of the bare VCSEL under a continuous injection current of 4 mA at room temperature, which reveal the single fundamental transverse mode operation. This is achieved by employing a small oxide aperture (~3 μm in diameter). **c** Calculated transmission amplitude and phase of the selected six meta-atoms at the wavelength of 978 nm. A complete 2π modulation range of propagation phase ($\varphi_u$ and $\varphi_v$) is achieved, keeping the phase difference between the two eigen-polarization channels fixed at: $|\triangle\varphi| = |\varphi_u - \varphi_v| = \pi$.

($J(x,y)$) of the integrated metasurface needs satisfying:

$$\begin{cases} J(x,y) \cdot |\sigma^-\rangle = e^{i\varphi_+(x,y)}|\sigma^+\rangle \\ J(x,y) \cdot |\sigma^+\rangle = e^{i\varphi_-(x,y)}|\sigma^-\rangle \end{cases} \qquad (3)$$

where $\varphi_+(x,y)$, and $\varphi_-(x,y)$ are two independently and arbitrarily modulated phase for the two spin states, respectively. This can be achieved by rewriting the Jones matrix through a linear combination of the geometric phase and propagation phase into the following form[39]:

$$J(x,y) = \begin{bmatrix} \frac{\exp[i\varphi_+(x,y)] + \exp[i\varphi_-(x,y)]}{2} & \frac{i\exp[i\varphi_-(x,y)] - i\exp[i\varphi_+(x,y)]}{2} \\ \frac{i\exp[i\varphi_-(x,y)] - i\exp[i\varphi_+(x,y)]}{2} & \frac{-\exp[i\varphi_+(x,y)] - \exp[i\varphi_-(x,y)]}{2} \end{bmatrix} \qquad (4)$$

$$\text{where} \begin{cases} \varphi_+(x,y) = \varphi_u + 2\theta, \\ \varphi_-(x,y) = \varphi_u - 2\theta, \\ |\varphi_u - \varphi_v| = \pi. \end{cases}$$ As the relations are valid for arbitrary $\varphi_u$

values, we can impose independent phase profiles to each of the spin states of a VCSEL, respectively.

To implement spin-decoupling of VCSEL emission with arbitrary phase profiles, we integrate Jones matrix metasurfaces composed of birefringent nano-fin arrays on the bottom-emitting facet of the laser by directly patterning the backside surface of the GaAs substrate, as illustrated in Fig. 1a. A very small oxide aperture (~3 μm in diameter) was employed to ensure the single-mode emission of the fabricated VCSEL, as confirmed in Fig. 1b, which helps to simplify the design of metasurface to address the well-defined fundamental transverse mode profiles of the incident beam. Note that a thick GaAs substrate (~630 μm) was intentionally used for the epitaxial growth of the laser

structure. The reason to use such a thick substrate is to significantly expand the laser beam after exiting from the laser cavity to a larger diameter of ~90 μm (considering a divergence of about 4° for light propagating in high index substrate). By doing this, we ensure sufficient diffraction and expansion of the laser beam to correctly manipulate the wavefront through the interaction with a large number of nanostructures forming the integrated metasurfaces. According to the size of the divergent beam, we defined larger metasurface of about 200 μm in diameter to avoid any diffraction of light caused by its physical boundary (see Supplementary Fig. S1 for details). For the design of metasurface, the complex transmission coefficients of the meta-atoms were calculated at the laser wavelength of 978 nm under normal incidence, as summarized in Fig. 1c. In addition, only small variations as a function of the incident angles (for changes up to 4° incidence) were noticed in Supplementary Fig. S2. Metasurfaces were designed for a precise monochromatic wavelength of the fabricated VCSELs, but due to the non-dispersive nature of the geometric phase[40] and the roughly constant phase intervals among the meta-atoms within the wavelength range from 974 nm to 984 nm (see Supplementary Figs. S3, S4), the Jones matrix requirement in Eq. 3 remains satisfied over the spectral range of interest. This property is beneficial to ensure that we could cover, with the same design, various emission frequencies of VCSELs in an array, mitigating any perturbation caused by the wavelength drifting, for example, due to the Ohmic heating, change of injection current as shown in Supplementary Fig. S5, and even multi-transverse modes emissions from VCSELs with a larger oxide aperture for high power operations. On the other hand, on-chip spin decoupling over the entire bandwidth could be further achieved by properly compensating the dispersive propagation phase

responses through rigorous dispersion engineering. For example, achromatic beam focusing/collimating requires the design of meta-surface not only to control the local phase response, but also to engineer both the group delay $\left(\frac{\partial\varphi}{\partial\omega}\right)$ and the group delay dispersion $\left(\frac{\partial^2\varphi}{\partial\omega^2}\right)$, respectively[12,33].

### Independent beam structuring of the spin-decoupled states of VCSEL

A simple rectangular pillar can work as an anisotropic effective trun-cated waveguide, allowing independent control of propagation phases imposed to two orthogonally polarized eigenmodes along its fast and slow axis by changing its width ($W$) and length ($L$), respectively, as depicted in the inset of Fig. 1a. For a complete $2\pi$ modulation of pro-pagation phase, six nanopillars numbered 1–6 with an equivalent phase step of $\pi/3$ were employed to build the meta-atom library in our design. Specifically, each unit cell was selected to impose $|\varphi_u - \varphi_v| = \pi$, whereas it exhibits a slightly different amplitude ($|T_u| \neq |T_v|$) between the two orthogonal polarizations (Fig. 1c), rendering it to operate as a half-wave plate with a moderate conversion efficiency in the range of 65–75% (see Supplementary Fig. S3). In this case, the rest of the unconverted impinging light, that is, the co-polarized beam, was not modulated by the PB phase, as explicitly expressed in Eq. 2, and thus was transmitted with uncontrolled phase.

In the first example, we exploit the integrated Jones matrix metasurface that composes of only a single layer of anisotropic structures to decouple and modify the two spin states of VCSEL independently. This feature is extremely relevant and could find applications in a spin injected VCSEL (spin-VCSEL) to establish the connection between the spin states of electrons with that of photons according to the injected spin electrons (see Discussion for details). To verify the capability of encoding arbitrary and independent phase functionality into each spin state of VCSEL, the wavefronts of both the output RCP and LCP components were holographically structured to display two different far-field images (a university logo and the letters "BJUT"), respectively, as illustrated in the inset of Fig. 2a. In experiment, a holographic Jones matrix metasurface was integrated with the VCSEL, of which the desired phase profiles were retrieved based on Gerchberg-Saxton algorithm to project two different targeted images

onto the white screen placed at $Z = 1\,cm$ from the device. Figure 2a demonstrates well-defined shape and sidewall profiles of the inte-grated metasurface, confirming the high accuracy of the developed nanofabrication process (see Method for fabrication details). Without the use of any polarization analyzer, two designed holographic images can be observed simultaneously, as shown in Fig. 2b. Additionally, a large beam spot was superimposed in the center of the images due to the existence of a co-polarized light presenting an uncontrolled phase profile, thus behaving as a diverging beam. The co-polarization com-ponent can be filtered out with a polarizer to unveil completely the holographic patterns, as shown in Fig. 2c. The polarization states (SOPs) of the two holographic images were further determined by adding additional polarizer and waveplate as CP polarization filter in the optical path (Fig. 2d, e), respectively, confirming the generation of two orthogonally polarized CP channels for spin-decoupled holo-graphic displays (see Supplementary Fig. S6 for measurement details).

The orthogonality of the VCSEL spin states can be utilized to further encrypt the images with polarization selectivity. For this pur-pose, another holographic VCSEL (referred as "Device A") was fabri-cated such that its CP components were designed to propagate along the same direction to mix their beam patterns, and the co-polarization light was later removed from the far-field image with a polarizer. (The top panel of Fig. 3a, b). Then, the individual holographic information can be accessed through CP filtering, respectively, as demonstrated in the top panel of Fig. 3c, d. The existence of co-polarization light caused by the non-unitary conversion efficiency of meta-atoms is utilized here as an additional degree to encrypt the optical information. The uncoded, but spatially overlaying, co-polarization signal can thus serve as an interference channel to hide the holographic information carried by the CP components. The concealed images can further be revealed through proper polarization filtering. This functionality can be better showcased by creating metasurfaces using meta-atoms with con-siderably reduced conversion efficiency to intentionally overlay a much stronger co-polarization beam spot with the holographic images (see Supplementary Fig. S7 for details). To this end, two new devices (referred as "Device B" and "Device C") were designed such that both the RCP components of device B and device C will display the same image of university logo. But we designed the phase of the LCP

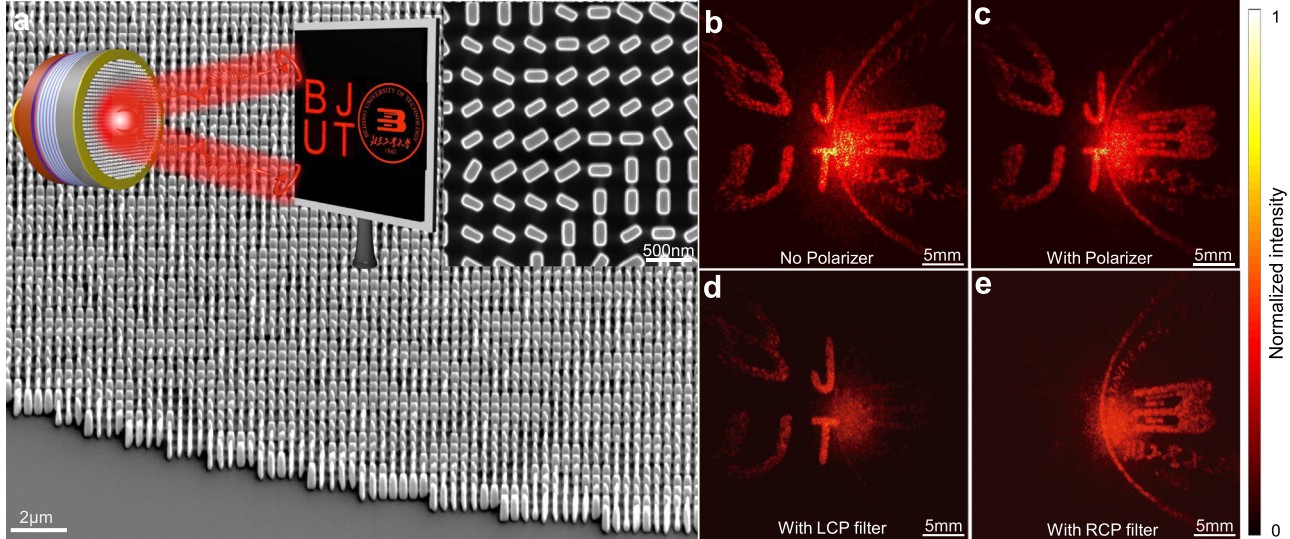

**Fig. 2 | Spin-decoupled holographic VCSEL. a** SEM images of the integrated holographic metasurface, revealing the high accuracy of the developed integration process. The inset schematically summarizes the example of dual-channel holo-graphic design to structure the far-field intensity profiles of the spin-decoupled VCSEL into two CP channels, in which two infrared images of the letters of "BJUT" and the university logo were directly projected onto a white screen for observation. **b, c** show the recorded far-field beam patterns without and with a polarizer, in which the uncontrolled co-polarization channel can be filtered out using a polar-izer. The SOPs of each spin-decoupled holographic channel can be further con-firmed upon the use of the polarization analyzers, respectively (**d, e**), which agrees well with the design.

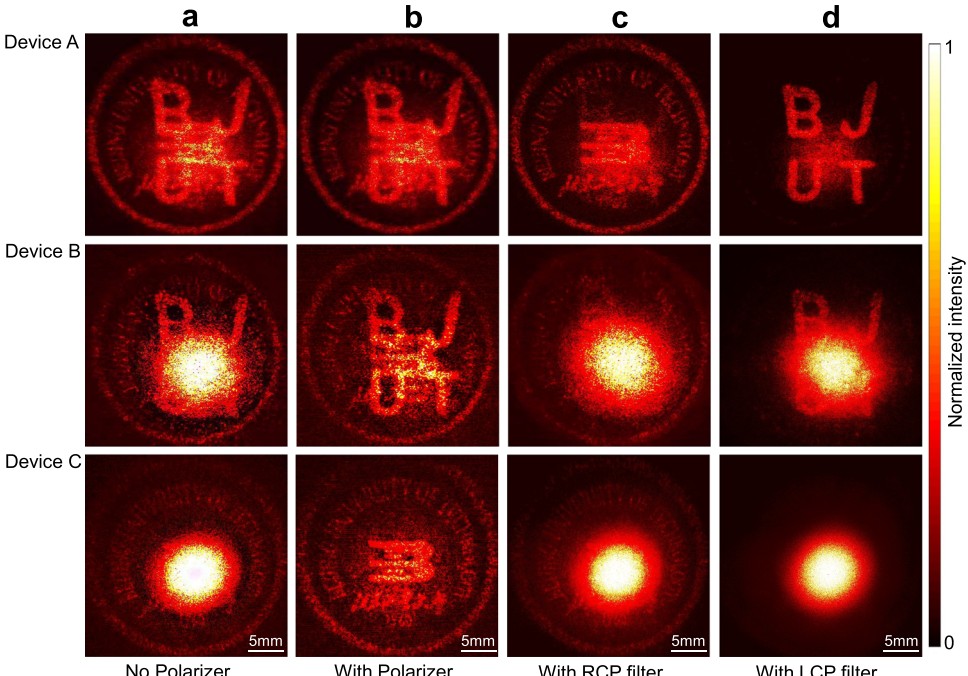

**Fig. 3 | Superposition of different beam profiles of the spin-decoupled holographic VCSEL, of which the polarization selectivity of each channel can be used for encryption purpose.** In this example, two sets of meta-atoms with different dimensions were employed to construct metasurfaces with relatively high efficiency (Device A) and low efficiency (Device B and Device C). Meanwhile, a broad diverging beam coming from the uncontrolled co-polarization channel was aligned with the spin-decoupled CPs beams to block the holographic images (**a**). Then, the divergent co-polarization beam spot can be removed from the holographic images with a polarizer, revealing the hidden information (**b**). Each spin state channel carrying the holographic information can be further selectively concealed by proper CP filtering, taking advantages of their orthogonality (**c**, **d**).

components differently for the two devices. Specifically, the LCP channel of device B was designed to display the letters of "BJUT" while that of the device C was kept divergent to leave this spin state unused for holographic encryption. Moreover, the encoded holographic images were resized in the design so that their central region in the image plane at $Z = 1$ cm will be superimposed, and thus blocked by the expanded beam spot of the strong co-polarized light, which are demonstrated in the middle and bottom panels of Fig. 3a. As suggested, the hidden holographic information encoded into the circular polarizations can be revealed with a polarizer (see the middle and bottom panels of Fig. 3b). Likewise, the individual spin state channel carrying the holographic pattern can be selectively concealed upon the use of CP filters, as shown in the middle and bottom panels of Fig. 3c, d, which is further illustrated in Supplementary Fig. S8.

Furthermore, such a dual-channel holographic laser chip with orthogonally polarized beams may also find applications in stereoscopic holographic display by assigning two images from different viewpoints of the same three-dimensional object[38]. In this way, a stereoscopic image can be viewed by further combining the beams with additional polarization components. Notably, compared with the widely reported metasurface holography, our integrated holographic VCSEL features a built-in light source without the requirement of external alignment, and therefore offers the advantages of compactness, low insertion loss, portability, and low cost. Moreover, the unique 2D array VCSELs provides a feasible and simple way to construct large size holographic images with high-resolution by stitching that of different individual laser pixels, rendering it a suitable platform to scale up the vectorial holographic display in contrast to the conventional metasurface holography.

## Generation of interference-free multi-polarization channels
As suggested by the Eq. 2, the divergent co-polarized light can be effectively depressed by employing half-wave plate meta-atoms with higher conversion efficiency, whereas it presents practical challenges related to nanofabrication imperfections and design tractability (see Supplementary Fig. S7 for details). Here we proposed a feasible and rather simpler solution to avoid both polarization mixing and poor image rendering in case of non-unitary PB efficiency. We demonstrated that by the proper design of Jones matrix metasurface, it is possible to generate interference-free multi-polarization channels output by spatially separating the optical path of each polarization channel. As a proof of concept, in the second example, we show that the co-polarization of VCSELs can be modulated as well by the same Jones matrix metasurface into collimated emission. Note that collimating the co-polarized light restricts its angular expansion, and thus suppresses its superposition with the deflected cross-polarized light in the projection plane, as illustrated in the inset of Fig. 4a. This is achieved by leveraging the fact that the propagation phase ($\varphi_u$) only relies on the material refractive index and the geometry of the meta-atoms, which therefore can exert equivalent phase modulation to both the co-polarized and cross-polarized light. Accordingly, the SEM images of the integrated Jones matrix metasurface are shown in Fig. 4a.

For this purpose, a gradient phase profile: $\phi_{deflector} = 2\pi - \frac{2\pi}{\lambda} y \sin\alpha$, where $\lambda$ represents the laser wavelength and $\alpha$ the deflection angle, is imposed to the RCP component using PB phase to deflect the RCP light along the transverse direction (we choose this to be $Y$-direction). By doing this, the LCP component will automatically gain an equal and opposite phase gradient, and thus be deflected at an angle of $-\alpha$ towards to $Y$-direction due to the conjugation phase functions of PB phase. In absence of propagation phase, the co-polarized light will remain unmodulated and should propagate straight as if there were no metasurface. To simultaneously collimate both the CPs beams and the co-polarized light, an additional hyperboloidal phase distribution: $\phi_{collimator}(x,y) = 2\pi - \frac{2\pi n}{\lambda}(\sqrt{x^2 + y^2 + f^2} - f)$, where $f$ is the focal length, $n$ is the refractive index of the substrate was introduced using

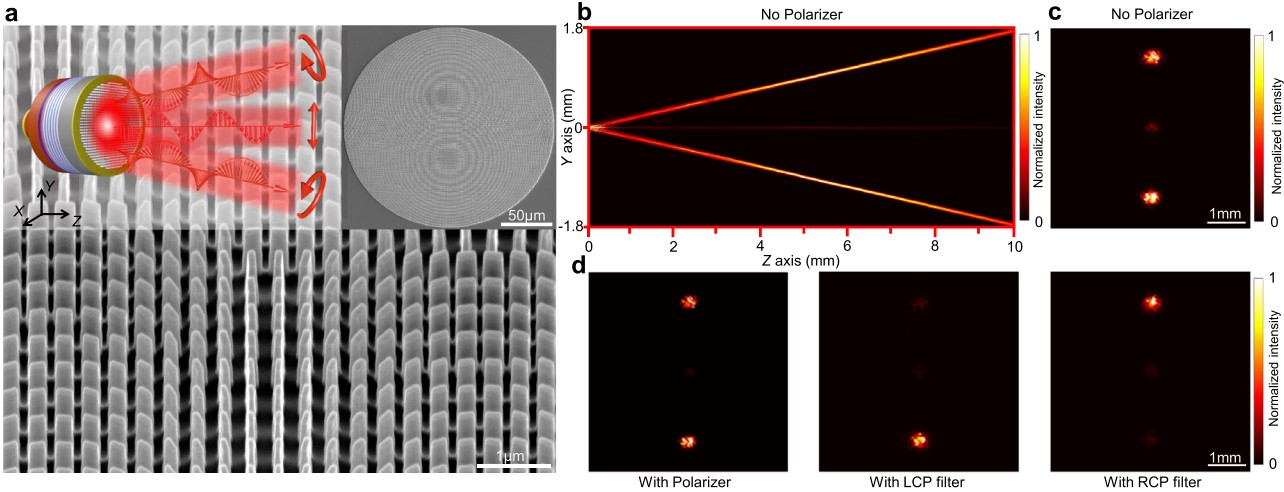

**Fig. 4 | Generation of interference-free multi-polarization channels. a** SEM images of the integrated Jones matrix metasurface (height 750 nm) for directional collimations. The inset is a schematic illustration of a three-channel collimated VCSEL emitting directional RCP, LCP, and LP beams enabled by metasurface integration. In this example, the cross-polarization components (RCP and LCP light) were modulated both by a deflecting PB phase profile and a collimating phase imposed by the propagation phase, while the co-polarization component was only modulated by the propagation phase. As a result, the cross-polarization components were respectively collimated and deflected at 10° (RCP) and −10° (LCP) toward to *Y*-axis, and the co-polarized light was collimated without deflection. The intensity beam profiles of the VCSEL were recorded along *Z* direction (**b**), and at *Z* = 1 cm (**c**) without polarizer, respectively, revealing the well separation of those three collimated beams without noticeable interferences along different directions. The SOPs of the generated beams were further determined using polarization analyzers, respectively (**d**).

propagation phase, which can compensate for the diffraction of both the cross-polarized and co-polarized components of VCSEL emissions. The resulting phase profiles for all three beams are summarized in Supplementary Fig. S9. Figure 4b, c show the measured beam intensity profiles of the VCSEL both along the propagation direction (*Z*-direction) and at *Z* = 1 cm, respectively, without the use of polarizer. As evidenced, three collimated beams with comparable profiles were well separated along different directions. To determine the SOPs of the generated beams, far-field emission patterns are analyzed at the distance of *Z* = 1 cm using additional polarizer and waveplate, respectively (see Fig. 4d and Supplementary Fig. S10). As expected, the combination of PB phase and propagation phase allows us to simultaneously collimate and separate of the cross-polarized and co-polarized channels of VCSEL. Therefore, this example provides a feasible approach to avoid the interferences between the otherwise diverging co-polarization channel and the cross-polarization channels by designing the integrated Jones matrix metasurface to spatially separate the optical paths of different polarization beams. The effectiveness of this simple design strategy was also confirmed in the case where a much stronger co-polarization beam exists due to the reduced conversion efficiency of the meta-atoms, as demonstrated in Supplementary Fig. S11. Furthermore, as a comparison, both the collimating phase and the deflecting phase were imposed to the RCP component of VCSEL by the integrated metasurface that deploys PB phase alone. In this case, the LCP beam will obtain a diverging phase due to the conjugate relationship with RCP light, while the co-polarization light will remain unmodulated and divergent. As a result, only the RCP component of the VCSEL is collimated by the metasurface, which manifests as a well-confined Gaussian beam according to the far-field measurement (see Supplementary Fig. S12).

The above design approaches of Jones matrix metasurface that rely on the propagation phase and geometric phase are shown to enable either (i) full decoupling of oppositely crossed CP beams, leaving the co-polarization channel uncontrolled, or (ii) collimating both co- and cross-polarizations, while the cross-polarizations are split into two conjugated circularly cross-polarized beams and modulated by the opposite PB phase profiles. Furthermore, by introducing another phase addressing mechanism as an additional degree of

freedom, it becomes possible to completely decouple all the CP channels of the Jones matrix metasurface with arbitrary modulated phases. For instance, Yuan et al. demonstrates that all the cross- and co-polarized outputs can be utilized and encoded independently by supplementing the metasurface design with chirality-assisted phase in addition to the propagation and geometric phases[41]. Therefore, integrating Jones matrix metasurface that decouples all the CP beams with VCSELs can serve as a generalized scheme to make the full utilization of CP components as independent data-carrying channels.

## On-chip manipulation of the polarization states of VCSELs
The fully decoupled spin states by the integrated Jones matrix metasurface provide a new pathway for on-chip manipulation of any polarization states of the VCSELs, as illustrated in Fig. 5a. To understand our proposition, we simply consider that any required polarization state $|\varphi\rangle$ can be expressed in the combination of the two opposite spin states according to:

$$|\varphi\rangle = A_+ \exp(i\varphi_+)|\sigma^+\rangle + A_- \exp(i\varphi_-)|\sigma^-\rangle \quad (5)$$

where $A_+$, $\varphi_+$, and $A_-$, $\varphi_-$ are the amplitude and phase term of the two spin states, respectively. Accordingly, the azimuth angle $\psi$ and ellipticity angle $\chi$ of $|\varphi\rangle$ can be derived as $\psi = \frac{\varphi_+ - \varphi_-}{2}$, and $\chi = \frac{1}{2}\arcsin\left(\frac{A_+^2 - A_-^2}{A_+^2 + A_-^2}\right)$. In this way, the SOPs of VCSELs can be fully controlled by tuning the phase retardation and amplitude ratio of the two spins states. Here, vector vortex beams were taken as examples to testify the feasibility of on-chip manipulating VCSEL SOPs. In the paraxial approximation, the SOPs of a monochromatic vector vortex beam can be represented by the linear combination of two spin states vortex beams of opposite topological charges[42,43]:

$$|E\rangle = \cos\left(\frac{\alpha}{2}\right)\exp\left(im\theta + i\frac{\beta}{2}\right)|\sigma^+\rangle + \sin\left(\frac{\alpha}{2}\right)\exp\left(in\theta - i\frac{\beta}{2}\right)|\sigma^-\rangle \quad (6)$$

where $\theta$ denotes the azimuthal angle in the polar coordinate system, $m$ and $n$ represent different topological charge numbers, $\cos(\frac{\alpha}{2})$ and $\sin(\frac{\alpha}{2})$

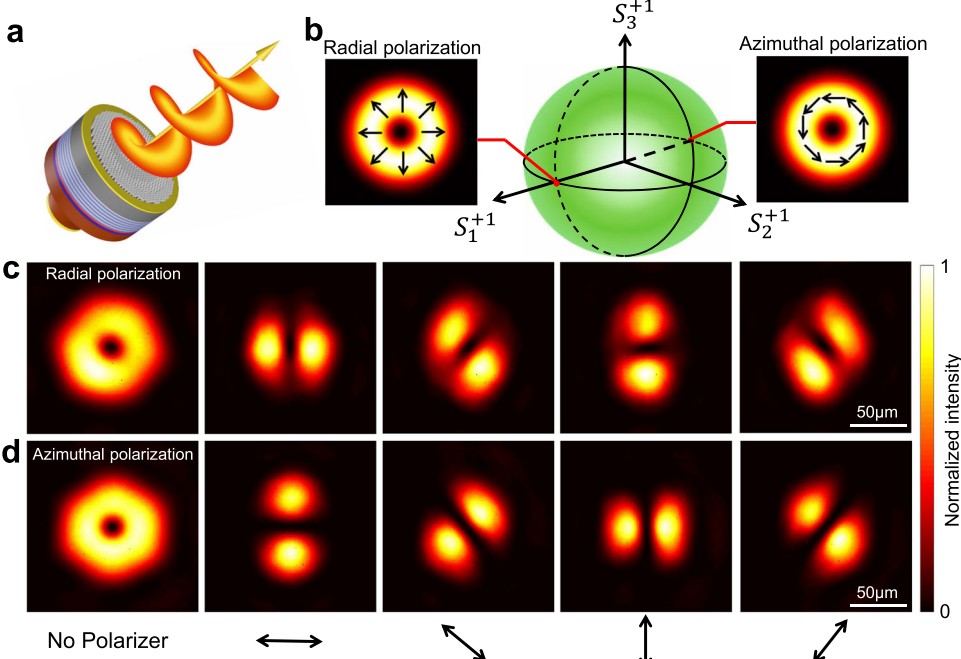

**Fig. 5 | On-chip manipulation of the polarization states of VCSELs. a** Schematic showing a metasurface integrated VCSEL that functions as a vector vortex laser. **b** As a proof of concept, two vector lasers with radical and azimuthal polarizations located on different positions of the HOPS are designed, respectively. The measured beam intensity profiles at $Z = 5$ cm in together with polarization analysis confirm the generations of radically polarized (**c**) and azimuthally polarized (**d**) vector beams, respectively. The black arrows indicate the orientation of the polarizer.

are the amplitude of $|\sigma^+\rangle$ and $|\sigma^-\rangle$, and $\beta$ is their relatively phase delay. By this means, a given vortex beam can be expressed by a point on the surface of higher-order Poincaré Sphere (HOPS) with coordinates $(\alpha,\beta)$, where $a \in [0,\pi], \beta \in [0,2\pi]$[44]. For simplicity and without loss of generality, two vector vortex beams, of which $\alpha = \frac{\pi}{2}$, polarization order $l = \frac{m-n}{2} = 1$, $\beta = 0$ and $\pi$, are designed and demonstrated, respectively, corresponding to two different positions on HOPS (radial and azimuthal polarizations), as depicted in Fig. 5b. In experiment, doughnut shape intensity profiles are observed from the two differently designed VCSELs (the first column in Fig. 5c, d). The spatially varying polarization profiles of both beams are revealed by placing a polarizer in front of the VCSEL. The images obtained by continuously rotating the polarizer create typical petal-shaped intensity distribution profiles, either parallel or orthogonal to the direction of polarizer (see Fig. 5c, d), revealing the generations of radially and azimuthally polarized vector beams with the polarization order equal to 1, respectively.

Despite that, in the above example, two orthogonal spin states with same amplitude are selected to create the vector vortex beams, it is worth noting that the Jones matrix metasurface is also capable to individually control the amplitude of each spin state[45]. Therefore, the on-chip integrated Jones matrix metasurface provides a powerful tool for the manipulation of VCSEL SOPs with full degree of freedoms.

## Discussion

Jones matrix metasurfaces enable the control of arbitrary and independent phase functions on any pair of orthogonal polarization states, adding a powerful tool to the toolbox of metasurface design. Capitalizing on this concept, a large variety of bi-functional metasurfaces have been previously demonstrated with a focus to multiplex the functionalities of metasurface by controlling the SOPs of the incident light in the free space. Moreover, such polarization dependent phase response can be used to sort the light of different SOPs, for example, operating as a separated polarimeter in combination of photo-detectors array[46]. With respect to those free-space applications, we would like to stress that the advantageous CMOS compatibility of Jones matrix metasurface can outperform the conventional polarization optics for on-chip integration applications, having profound potential to upgrade conventional semiconductor optoelectronic devices with spin-decoupled characteristics.

In addition to controlling the wavefront of the laser, the metasurface integration can be also used to modify the laser mode and dynamic. In this regard, impressive controls over the orbital angular momentum[47,48] and chirality of the light[49] have been proposed and demonstrated by directly introducing metasurface inside the laser cavity. Nevertheless, there are several critical challenges, including the deterioration of material optical quality caused by the nanofabrication, the effective injection of carriers under electrical pumping, that need to be solved to realize electrically injected metasurface lasers. In comparison, integrating metasurfaces on the emitting facet of the laser has the advantages of high compatibilities with standard laser process, such as manufacturing, packaging, electrical injection solutions, etc. Therefore, there would be substantial interest to further exploit the metasurface integrated outside the laser cavity for on-chip control of lasing characteristics, while preserving the intact laser structure. To this end, one of the potential strategies could be coupling the reflection of the metasurface into the laser cavity for mode selection by properly designing the reflectivity of both the metasurface and the DBRs mirror.

Furthermore, we envision that the integration of spin-decoupled metasurface could provide an ideal interface to manipulate and read-out the spin-polarized photons emitted by the spin-VCSEL. This will find particular interest in developing high-performance spin-optoelectronic devices, in which the spin momentums of both injected carriers and emitted photons are directly connected for the generation of spin-controlled emissions. For instance, despite the spin amplification in the spin-VCSEL allows generating CP light with high degree of polarization (DOP) even under the injection of low polarized carriers[50], the current DOP of a spin-VCSEL is limited due to the fast spin flipping and thermal quenching rate in the semiconductor quantum wells (QWs). Hence, integration of spin-decoupled metasurface can be used

not only to facilitate chiral emissions with high DOP by filtering out the unwanted CP light, but also to encode both two spin components independently. On the other hand, combing the design of spin-decoupled metasurface with various helicity-switching mechanisms of spin-VCSELs, such as by controlling the applied optical[51], electrical[52], or magnetic fields[53], will give rise to a new type of tunable laser with spin controlled reconfigurable beam functionalities. In addition to shaping the emitted spin photons, the electron spins of the VCSEL can also be controlled using the metasuface in the sense that spatial distributions of the optically injected spin carriers can be manipulated by independently guiding the CP pumping beams of different chirality into specifically assigned regions in the active medium. In this way, spatially selective generation of spin electrons can be implemented inside different areas of semiconductor QWs.

In conclusion, by monolithically integrating Jones matrix metasurfaces with commercially available VCSELs, this work proposes and proves that the optical spin states of VCSELs can be decoupled with complete phase modulation in an ultracompact on-chip manner. Our approach provides a timely opportunity to access the optical spin states of VCSELs with the capability to independently manipulate their phase functions. To demonstrate this concept, spin-decoupled VCSELs with various functionalities were designed and presented, including chirality selective dual-channel holographic displays, and directional generation of multi-collimated beams with polarization dependence. Moreover, we show that using the spin-decoupled CP beams as the basis any arbitrary polarization states can be explicitly generated from VCSELs. As an example, vector beams with radial and azimuthal polarizations were created by judiciously superimposing the spin-decoupled beam components with designed phase relations, respectively. Therefore, our finding can serve as a generalized scheme for the wavefront structuring of VCSELs with complete controllability over the amplitude, phase, and polarization states in an integrated way.

## Methods

### Numerical simulations

The complex transmission coefficients of the meta-atoms were obtained numerically using full-wave finite-difference time-domain (FDTD) simulations. For the FDTD simulations, rectangular GaAs nanopillars are arranged on a GaAs bulk substrate in a subwavelength lattice constant of 360 nm. Plane wave polarized along $X$- and $Y$- axes are employed as the excitation sources, respectively, in which periodic boundary conditions are applied along all the in-plane directions and perfectly matched layer (PML) boundary condition is used in the direction of the light propagation. The phase shifts ($\varphi_u$ and $\varphi_v$), and amplitude ($T_u$ and $T_v$) are obtained by parameter sweeping of the $W$ and $L$ of the GaAs nanopillars while keeps the rotation angle ($\theta$) at zero, as depicted in the inset of Fig. 1a.

### On-chip integration of metasurface

Prior to the integration of metasurface, a commercially available VCSEL wafer was processed into the backside emitting configuration (see Supplementary Fig. S1 for fabrication details). After that, metasurfaces were integrated at the backside surface of the GaAs substrate by directly structuring the substrate into nanopillars. To this end, a 180 nm thick Hydrogen silsesquioxane (HSQ) layer was spin-coated onto the substrate and defined into the metasurface patterns using electron beam lithography (EBL), serving as the hard mask. To further transfer the defined metasurface patterns into the GaAs substrate, inductively coupled plasma reaction ion etching (ICP-RIE) etching was used with optimized conditions to minimize the surface damage. Due to the lack of an etch-stopping layer in the current design of the laser structure, the etching depth was adjusted by controlling the etching time.

## Reporting summary

Further information on research design is available in the Nature Portfolio Reporting Summary linked to this article.

## Data availability

All data used in this study are available from the corresponding authors upon reasonable request.

## Code availability

The code used for the meta-hologram design is available from the corresponding authors upon reasonable request.

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

## Acknowledgements

P.F., C.X. and Y.X. acknowledge support from the National Key R&D Program of China (2018YFA0209000), the National Natural Science Foundation of China (62074011, 61874145, 62134008), Beijing Nova Program (Z201100006820096). P.N., and P.G. acknowledge support from European Research Council (ERC) under the European Union's Horizon 2020 research and innovation program (grant agreement FLATLIGHT No 639109). The authors thank the Nanofabrication Laboratory at National Center for Nanoscience and Technology for sample fabrication.

## Author contributions

P.N., Y.X., and P.G. conceived the idea and coordinated the experiment. P.C., C.X., Y.X. and P.G. supervised the project. P.N. performed the design and numerical simulations. P.F. carried out the fabrication and measurement. P.N., Y.X., and P.G. wrote the paper draft. All authors participated in improving the final version of the paper.

## Competing interests

The authors declare no competing interests.
