## [Peer Review File · Nature Communications]

Reviewers' Comments:

Reviewer #1:

Remarks to the Author:

The authors propose the integration of Jones matrix metasurfaces with VCSELs to achieve spin-decoupling of VCSEL emissions. The integration enables the arbitrary modulation of wavefront of the VCSEL emission in each spin state. As a demonstration, the authors experimentally realize dual-channel holographic image projection and directional generation of multi-collimated beams with polarization dependence. Even though this paper does not present new physics in terms of metasurface design, the integration of state-of-art metasurface with the commercially available product (VCSEL) is impressive as suggesting a new type of spin-optoelectronic device. The manuscript is well-written in general, and the experimental demonstration supports the proposed concept and design throughout the paper. Thus, I recommend this paper to be published after the following minor comments are addressed.

Comments:

1. In line 155, the author pointed out the eq. 3 can be satisfied for a sufficiently wide range of wavelength due to the relatively broadband response of the meta-atom with the ref. 40. However, I do not think this is the case when the propagation phase is utilized for arbitrary wavefront modulation at each orthogonal spin state. In the previous study (ref. 40), the authors use only geometric phase to remove chromatic dispersion that matters in the propagation phase. From the fact that the dispersive propagation phase is used here, it does not seem that Eq. 3. is met over a wide range of wavelengths in this concept.

1-1. it would be helpful for the readers to suggest how to design or compensate for such a chromatically dispersive response of the spin-decoupled Jones matrix metasurface while maintaining its spin-independent beam structuring function.

1-2. Please suggest the conversion efficiency of the nanopillars (the six-unit cells) over a wavelength range of interest to support the authors' claim.

2. I wondered if the divergent beam emitted from VCSEL chip lead to wrong or incorrect phase delay in the metaatoms. The beam emitted from the small oxide aperture is expanded to a larger diameter of $\sim 90\mu\text{m}$ while propagating through the thick substrate. Therefore, the beam is divergent with some angle; in other words, the beam will be obliquely incident on the metasurface. Especially, the angle of incidence becomes larger in the metaatoms located in the edge part compared to the ones in the center. Could the authors describe the phase delay response of the metaatoms under oblique incidence and how this affects to the overall performance of on-chip integrated metasurface?

Reviewer #2:

Remarks to the Author:

The manuscript described a spin-decoupled VCSEL device where a Jones matrix metasurface is directly embedded on one facet of the laser. By designing the metasurface polarization response, the authors demonstrated dual-channel holograms, polarization-dependent beam splitting, and generation of higher order Poincare states. Compared to previous metasurface work, this work contains significant novelty in that the metasurface is directly integrated onto the laser, making the overall optical system ultra-compact and self-aligned, and opening up new opportunities in spin-optoelectronics. The manuscript is well written. The overall quality of the work is high and of interest to a broad community. I recommend the publication of the manuscript if the authors can address the following questions:

There is a very strong zeroth order light in the experiment. The authors mentioned that it is due to fabrication imperfection, and proposed to 1) use it as a feature for encryption; and/or 2) separate it spatially from the other polarization channels. While those are neat ideas, however, in many applications ideally one would like to get rid of the zeroth order as much as possible. Therefore, I think it is important that the authors put more rigorous and in-depth analysis on the possible rootcauses and methods to mitigate/suppress the zeroth order. For example, what aspect of

fabrication imperfection is the leading cause of low efficiency? Is it pillar height, width, material index, or something else? What are the tolerances for each of them? Taking into account the realistic fabrication conditions, what efficiency can one reasonably expect to have?

In this work, the metasurface is used to control the wavefront after light is emitted from the laser. Is it possible to include the metasurface inside the laser cavity and use it to directly modify the laser mode or dynamics?

The authors mentioned spin-optoelectronics as one area of potential applications. However, there is no concrete example or explanation of how such a spin-decoupled VCSEL device could be used to control the spin of electrons. I suggest the authors to provide at least one concrete example of application and describe in more details.

In this response letter, we provide a detailed point-by-point reply to each of the reviewers' comments on the manuscript (NCOMMS-22-26135). Our responses are shown in blue, and the revised texts in the main manuscript are marked in red.

REVIEWERS' COMMENTS

Reviewer #1:

The authors propose the integration of Jones matrix metasurfaces with VCSELs to achieve spin-decoupling of VCSEL emissions. The integration enables the arbitrary modulation of wavefront of the VCSEL emission in each spin state. As a demonstration, the authors experimentally realize dual-channel holographic image projection and directional generation of multi-collimated beams with polarization dependence. Even though this paper does not present new physics in terms of metasurface design, the integration of state-of-art metasurface with the commercially available product (VCSEL) is impressive as suggesting a new type of spin-optoelectronic device. The manuscript is well-written in general, and the experimental demonstration supports the proposed concept and design throughout the paper. Thus, I recommend this paper to be published after the following minor comments are addressed.

Response:

We thank the reviewer for highly recognizing our work and for their recommendation to publish our manuscript. We have carefully studied all the comments and addressed the points raised by the reviewer as below:

Comments:

1. In line 155, the author pointed out the eq. 3 can be satisfied for a sufficiently wide range of wavelength due to the relatively broadband response of the meta-atom with the ref. 40. However, I do not think this is the case when the propagation phase is utilized for arbitrary wavefront modulation at each orthogonal spin state. In the previous study (ref. 40), the authors use only geometric phase to remove chromatic dispersion that matters in the propagation phase. From the fact that the dispersive propagation phase is used here, it does not seem that Eq. 3. is met over a wide range of wavelengths in this concept.

Response:

We thank the reviewer for pointing out the ambiguity of this sentence. In addition to considering the non-dispersive nature of the geometric phase, the propagation phases of the meta-atoms were also found to change only slightly as the wavelength changes from 974 nm to 984 nm, as revealed below in Fig. S4. This is due to the large and relatively constant index of GaAs in this spectral range. More importantly, it is worth noting that the intervals of the phase steps provided by those six elements remains the same over this wavelength region, which keeps the phase discretization conditions unchanged. Therefore, it ensures that the beam structuring properties of the spin-decoupled metasurfaces in the current design would be sufficiently robust against the wavelength drifting of the emitted light from the VCSEL within this spectral range, for example, due to the change of

injected currents, as indicated in Fig. S5 Supplementary Information. Meanwhile, we agree with the reviewer that a small spectrum span of around 10 nm, corresponding to the emission wavelength shifts of the fabricated VCSEL, studied here does not prevail the term of “relatively broadband response” in the previous sentence. Therefore, following this comment, we have clarified this part both in the main text of the revised manuscript and in Fig. S4 Supplementary information, as summarized below:

“Metasurfaces were designed for a precise monochromatic wavelength of the fabricated VCSELs, but due to the non-dispersive nature of the geometric phase⁴⁰ and the constant phase intervals among the meta-atoms within the wavelength range from 974 nm to 984 nm, the Jones matrix requirement in Eq. 3 remains satisfied over the spectral range of interest, as revealed in Fig. S3 and S4.”

Fig. S4 Simulated propagation phase delay φ_u (a) and φ_v (b) of the selected six meta-atoms within the wavelength range of interest from 974 nm to 984 nm, respectively.

1-1. it would be helpful for the readers to suggest how to design or compensate for such a chromatically dispersive response of the spin-decoupled Jones matrix metasurface while maintaining its spin-independent beam structuring function.

Response:

We thank the reviewer for this suggestion. The general approach to compensate the dispersive phase response of the propagation phase relies on rigorous dispersion engineering over the entire bandwidth, which would allow achieving fully non-dispersive beam structuring function for extra broadband spin-decoupling applications. For this purpose, it requires the design of metasurface not only to control the local phase response, but also to engineer both the group delay ($\frac{\partial\varphi}{\partial\omega}$) and the group delay dispersion ($\frac{\partial^2\varphi}{\partial\omega^2}$), respectively.

Following this suggestion, we have added the discussion on the design principle to compensate the dispersion phase response of the spin-decoupled metasurface for ultra-broadband applications in the revised manuscript, as below:

“On the other hand, on-chip spin decoupling over the entire bandwidth could be further achieved by properly compensating the dispersive propagation phase responses through rigorous dispersion engineering. For example, achromatic beam focusing/collimating requires the design of metasurface

not only to control the local phase response, but also to engineer both the group delay ($\frac{\partial \varphi}{\partial \omega}$) and the group delay dispersion ($\frac{\partial^2 \varphi}{\partial \omega^2}$), respectively.^{12,33}

1-2. Please suggest the conversion efficiency of the nanopillars (the six-unit cells) over a wavelength range of interest to support the authors' claim.

Response:

We thank the reviewer for this suggestion. Following this advice, the conversion efficiency of the six meta-atoms over the lasing wavelength range from 974 nm to 984 nm have been studied and added as Fig. S3 in the Supplementary Information. It shows that the selected six meta-elements exhibit rather stable conversion efficiency within this spectral range.

Fig. S3 Simulated conversion efficiency of the selected six meta-atoms within the wavelength range of interest from 974 nm to 984 nm.

2. I wondered if the divergent beam emitted from VCSEL chip lead to wrong or incorrect phase delay in the metaatoms. The beam emitted from the small oxide aperture is expanded to a larger diameter of ~ 90um while propagating through the thick substrate. Therefore, the beam is divergent with some angle; in other words, the beam will be obliquely incident on the metasurface. Especially, the angle of incidence becomes larger in the metaatoms located in the edge part compared to the ones in the center. Could the authors describe the phase delay response of the metaatoms under oblique incidence and how this affects to the overall performance of on-chip integrated metasurface?

Response:

We thank the reviewer for bringing up this important point. Yes, considering the beam divergence of the VCSEL emission, the change of the incident angle upon the meta-atoms away from the center is one of the potential origins of phase errors that needs to be considered in the design of metasurface integration. In our laser structure, the maximum incident angle (θ) that corresponds to the meta-atoms located at the edge of the beam, as depicted in Fig. S2a, was estimated to be $\sim 4^\circ$ according to the relation: $\theta = \tan^{-1}(\frac{D}{2f})$, where D represents the diameter of the incident beam and f is the distance between the light emitting aperture and the metasurface. Accordingly, the phase response

of each meta-atoms under different oblique incidence up to 4° was calculated and compared, as summarized in Fig. S2b and 2c. It is found that the phase response exhibits only small variations within this small range of incident angles. Therefore, the main cause for the phase errors in the current design is attributed to the small number of phase discretization levels. And we expect that the overall performance of the integrated metasurface can be further improved by mitigating the phase errors in the design, by both increasing the number of the phase levels and compensating the slight angle dependance of the phase responses.

Fig. S2 (a) Schematic illustration of the angle dependence of the incident beam upon the meta-atoms; Simulated propagation phase delay ϕ_u (b) and ϕ_v (c) of the selected six meta-atoms at a laser emission wavelength of 978 nm for different incident angles.

Following this suggestion, we have discussed more details on both the angle dependent phase response of the meta-atoms and the potential strategies to further mitigate the overall phase errors in the metasurface design both in the revised manuscript and as Fig. S2 in the Supplementary Information:

“In addition, only small variations as a function of the incident angles (for changes up to 4° incidence) were noticed in Fig. S2.”

Reviewer #2:

The manuscript described a spin-decoupled VCSEL device where a Jones matrix metasurface is directly embedded on one facet of the laser. By designing the metasurface polarization response, the authors demonstrated dual-channel holograms, polarization-dependent beam splitting, and generation of higher order Poincaré states. Compared to previous metasurface work, this work contains significant novelty in that the metasurface is directly integrated onto the laser, making the overall optical system ultra-compact and self-aligned, and opening up new opportunities in spin-optoelectronics. The manuscript is well written. The overall quality of the work is high and of interest to a broad community. I recommend the publication of the manuscript if the authors can address the following questions:

We thank the reviewer for highly recognizing our work and for their recommendation to publish our manuscript. We have carefully studied all the comments and addressed the points raised by the reviewer as below:

Comments:

1. There is a very strong zeroth order light in the experiment. The authors mentioned that it is due to fabrication imperfection, and proposed to 1) use it as a feature for encryption; and/or 2) separate it spatially from the other polarization channels. While those are neat ideas, however, in many applications ideally one would like to get rid of the zeroth order as much as possible. Therefore, I think it is important that the authors put more rigorous and in-depth analysis on the possible root causes and methods to mitigate/suppress the zeroth order. For example, what aspect of fabrication imperfection is the leading cause of low efficiency? Is it pillar height, width, material index, or something else? What are the tolerances for each of them? Taking into account the realistic fabrication conditions, what efficiency can one reasonably expect to have?

Response:

We thank the reviewer for bringing up this important point and for the advice. The strong zeroth order light observed in initial work has been considerably suppressed to now reach high-efficiency operation by further improving both the fabrication and design in the following ways:

- 1) Regarding the fabrication imperfection, the major manufacturing error in our work is attributed to the variation of the etching depth among the nano-pillars with varying gap width, as revealed below in Fig. 1b. This will cause substantial deviations of the device actual performance from the design, due to both the phase and transmission errors, especially for those meta-atoms with smaller gaps. To address this problem, we suggest that the etching depth deviations can be mitigated by choosing the meta-atoms with larger separation distance. To prove this, we increase the overall thickness of the metasurface to 750 nm, which allows us to select new meta-atoms with smaller dimensions to achieve the required phase modulations while keeping the same lattice constant (Fig. 1c). In this way, new meta-devices were fabricated, which exhibit considerably improved uniformity of the etching depth, as confirmed in Fig. 1d. We also note that the new meta-atoms (height 750 nm) could provide higher conversion efficiency than the previous ones (height 700 nm), according to the numerical simulations, as shown in Fig. 1e. By doing this, we experimentally demonstrated that the zero-order light can be significantly suppressed, and the overall performance has been significantly improved in the new devices, as shown in Fig. 2.

Moreover, to solve the problematic etch depth variations of the nanostructures thoroughly, we also suggest that a rather rigorous solution can be implemented by introducing an etching stopping layer on the backside surface of the substrate through an additional layer of GaAs with well-defined thickness. In this way, it will allow precisely controlling the etching depth with ultra-high resolution even in the sub-nanometer scale by taking the advantage of the epitaxy technology.

Fig. 1 Increasing the height of the meta-atoms from 700 nm to 750 nm allows selecting meta-atoms with smaller dimensions, as shown in the SEM images (a-d), while satisfying the required phase modulation coverage with the same lattice constant. (b) shows that the larger dimensions of the 700-nm-tall meta-atoms causes larger variations of the etching depth among the nano-pillars with varying gap width. (e) compared the conversion efficiency of the meta-atoms with different height.

Fig. 2 the measured far-field intensity patterns of the spin-decoupled holographic VCSELs with different height. It reveals that the metasurfaces composed of 750 nm-tall meta-atoms (Right panel) exhibit better performance with significantly suppressed zero-order light than the ones with 700 nm height. (Left panel)

- 2) The efficiency of the device is also restricted by the relatively small number of phase discretization levels ($N = 6$) employed in the current design, which caused substantial phase errors. According to the diffraction theory, the diffraction efficiency of the structure with a finite number (N) of phase discretization levels can be given as $[\frac{N}{\pi} \sin(\frac{\pi}{N})]^2$. [1, 2] This equation indicates that the diffraction efficiency could be significantly increased by increasing the number of phase discretization levels, for example, from about 91% ($N = 6$)

to about 98.7% ($N=16$). Therefore, the overall performance of the device can be further improved by deploying more phase levels to construct the metasurface.

Regarding the achievable efficiency, a high efficiency larger than 80% can be expected from the proposed devices, taking into account the realistic fabrication conditions. This conclusion is based on the fact that high performance metasurface hologram with impressive efficiency $\sim 80\%$ has been experimentally demonstrated using plasmonic meta-atoms with 16 phase levels by Zheng *et al.* [3] It is worth noting that the dielectric nature of the meta-atoms employed in this work has the advantage to avoid the problematic Ohmic losses associated with their plasmonic counterpart. Therefore, we are expecting that a much more efficient spin decoupled VCSEL can be realized by further optimizing both the design and fabrication.

Following this suggestion, we have added in-depth discussion on different approaches to further suppress the zero-order light for high efficiency operations in the Supplementary Information, as below:

Fig. S7 The selected two sets of meta-atoms to construct metasurfaces with different conversion efficiency. Increasing the height of the meta-atoms from 700 nm to 750 nm allows us to select meta-atoms with smaller dimensions, as shown in (a-d), to achieve the required phase modulation coverage while keeping the same lattice constant. It is worth noting that although the meta-atoms with different height only exhibit slight difference in conversion efficiency according to (e), the larger dimensions of the 700-nm-tall meta-atoms caused more significant fabrication errors, mainly manifesting as larger variations of the etching depth among the nano-pillars with varying gap width, as revealed in (b). This will cause substantial deviations of the device actual performance from the design, due to both the phase and transmission errors, especially for those meta-atoms with smaller separation distance. As a result, the fabricated metasurfaces with 700 nm height present a much stronger co-polarization beam spot, as observed in both the device B and device C. On the other hand, to further increase the performance of metasurface for highly efficient operations, precise control of the etching depth of meta-atoms with different dimensions can be achieved by introducing an etching stopping layer on the backside surface of the substrate through an additional layer of GaAs with well-defined thickness, which can provide ultrahigh resolution even in the sub-nanometer scale by taking the advantage of the epitaxy technology. Moreover, the overall performance of the device can be further enhanced by employing more phase levels to construct the metasurface to mitigate the substantial phase errors due to the relatively small number of phase discretization levels ($N=6$) in the current design.

1. E. Hasman, V. Kleiner, G. Biener, and A. Niv, "Polarization dependent focusing lens by use of

- quantized Pancharatnam–Berry phase diffractive optics," *Applied physics letters* **82**, 328-330 (2003).
2. X. Luo, F. Zhang, M. Pu, Y. Guo, X. Li, and X. Ma, "Recent advances of wide-angle metalenses: principle, design, and applications," *Nanophotonics* **11**, 1-20 (2022).
 3. G. Zheng, H. Mühlenbernd, M. Kenney, G. Li, T. Zentgraf, and S. Zhang, "Metasurface holograms reaching 80% efficiency," *Nature nanotechnology* **10**, 308-312 (2015).

2. In this work, the metasurface is used to control the wavefront after light is emitted from the laser. Is it possible to include the metasurface inside the laser cavity and use it to directly modify the laser mode or dynamics?

Response:

We thank the reviewer for bringing up this excellent point. Yes, introducing the metasurface inside the laser cavity can be used as a compact approach to directly modify the laser mode and dynamic at source, which has the advantages of reducing the number of optical elements and the cavity complexity. For example, Sroor *et al* generated ultra-high order ($l = 100$) of orbital angular momentum mode with high purity from an optically pumped solid-state laser by placing the metasurface between the cavity mirrors. [4] However, due to the intrinsic compactness of the semiconductor lasers and the ultra-sensitivity of the semiconductor quantum wells to either the material defects or changes in environment, it is facing great challenges to directly fabricate the metasurface inside the cavity of a semiconductor laser without impairing the laser performance, such as the deterioration of the quantum efficiency, the threshold behavior, or the beam quality, especially for the case of an electrically pumped laser. For example, Ha *et al* demonstrated directional lasing by fabricating the semiconductor active layer into two-dimensional array of resonant nanoantenna, but their device can only operate under the optical excitation at cryogenic temperature up to 200 K due to the low optical gain caused by the nanofabrication. [5] In addition, since the light will undergo multiple passes backwards and forwards through the metasurface integrated inside the cavity, any imperfection of the metasurface will be dramatically amplified by the cavity effect.

In comparison, we would like to highlight that the non-intrusive nature of integrating the metasurface outside the laser cavity drastically simplifies the design and integration of beam shaping metasurface and prevents from interfering with the lasing characteristics. This makes our approach fully compatible with state-of-the-art VCSELs technologies, including the wafer-level fabrication process, the standard packaging processes, electrical injection solutions and theoretical analysis. Those merits have the advantages for practical applications.

We also would like to suggest that although the integrated metasurface in our approach is not spatially coupled to the laser cavity, it is also possible to use such metasurface to influence the laser mode and dynamics. For example, optical feedback can be sent into the laser cavity in the form of reflected light by judiciously increasing the reflectivity of the metasurface and reducing the reflectivity of the DBRs mirror, which would allow on-chip control of lasing characteristics while preserving the intact laser structure.

Following this comment, both the pros and cons of the two different approaches to integrate metasurface inside and outside the laser cavity have been discussed and compared, respectively, in the discussion section of the revised manuscript, as below:

“It is worth noting that in addition to controlling the wavefront of the laser, the metasurface

integration can be also used to modify the laser mode and dynamic. In this regard, impressive controls over the orbital angular momentum^{46,47} and chirality of the light⁴⁸ have been proposed and demonstrated by directly introducing metasurface inside the laser cavity. Nevertheless, there are several challenging issues, including the deterioration of material optical quality caused by the nanofabrication, the effective injection of carriers by electrical pumping, that need to be solved to realize electrically injected metasurface lasers. In comparison, integrating metasurfaces on the emitting facet of the laser has the advantages of high compatibilities with standard laser processes, such as manufacturing, packaging, electrical injection solutions, *etc.* Therefore, there would be substantial interest to further exploit the metasurface integrated outside the laser cavity for on-chip control of lasing characteristics while preserving the intact laser structure. To this end, one of the potential strategies could be coupling the reflection component of the metasurface into the laser cavity for mode selection by judiciously designing the reflectivity of both the metasurface and the DBRs mirror.”

4. H. Sroor, Y.-W. Huang, B. Sephton, D. Naidoo, A. Vallés, V. Ginis, C.-W. Qiu, A. Ambrosio, F. Capasso, and A. Forbes, "High-purity orbital angular momentum states from a visible metasurface laser," *Nature Photonics* **14**, 498-503 (2020).

5. S. T. Ha, Y. H. Fu, N. K. Emani, Z. Pan, R. M. Bakker, R. Paniagua-Domínguez, and A. I. Kuznetsov, "Directional lasing in resonant semiconductor nanoantenna arrays," *Nature nanotechnology* **13**, 1042-1047 (2018).

3. The authors mentioned spin-optoelectronics as one area of potential applications. However, there is no concrete example or explanation of how such a spin-decoupled VCSEL device could be used to control the spin of electrons. I suggest the authors to provide at least one concrete example of application and describe in more details.

Response:

We are thankful to the reviewer for this valuable advice. Compared to the remarkable breakthroughs in developing spin polarized light sources, such as the developments of chiral cavities and metasurfaces without time-reversal symmetry to break the balance between left and right circular polarization, [6, 7] and the very recent demonstration of chiral emission from resonant metasurface, [8] research on the integration of spin-decouple optics with the emerging spin-optoelectronic devices is behind. To the best of our knowledge, there is no report today to spin-decouple the state-of-the-art VCSELs with complete phase control. We envision that the proposed on-chip spin decoupled VCSELs could potentially enrich the applications of spin-optoelectronic devices in the following aspects:

- 1, Despite that spin amplification has been widely demonstrated in spin injected VCSEL, which allows generating circularly polarized light with high degree of polarization (DOP) even by injecting low polarized spin carriers into the active medium, [9] the achievable DOP of spin-VCSEL is currently still far below 100% due to the fast spin flipping rate and thermal quenching rate in the semiconductor quantum wells (QWs). In this regard, an impressive high DOP exceeding 90% was recently demonstrated in Ref [10]. Therefore, integration of spin-decoupled metasurface provides a timely solution not only to ensure fully spin polarized emissions by filtering out the unwanted circular polarization light, but also to independently functionalize the two spin polarization channels

as demonstrated in this work.

2, Helicity switching of spin-VCSELs has been proposed and successfully realized, such as by controlling the optical excitation, [11] the electrical injection, [12] or the magnetic field. [13] Therefore, combing the spin decoupled VCSEL with different helicity switching mechanisms allows the implementation of spin controlled light emissions with reconfigurable beam functionalities, which will give rise to a new type of tunable laser.

3, In addition to beam shaping the spin polarized light emissions, controlling the spins of electrons can be realized in the sense that the spatial distributions of optically injected spin carriers can be manipulated by independently guiding the spin polarized pumping beams of different chirality into specifically assigned regions of the active medium. In this way, it will allow spatially selective generation of spin electrons inside different areas of semiconductor QWs.

Following this suggestion, we have added discussion with more details on the conceptual applications of the spin decouple VCSEL in the revised manuscript as below:

“Furthermore, we envision that the integration of spin-decoupled metasurface would serve as an ideal interface to manipulate and read-out the spin polarized photons emitted by a spin injected VCSEL (spin-VCSEL). This could find particular interest in developing high-performance spin-optoelectronic devices, in which the spin momentums of both injected carriers and emitted photons are directly connected for the generation of spin-controlled chiral emissions. For instance, despite the spin amplification in the spin-VCSEL allows generating CP light with high degree of polarization (DOP) even under the injection of low polarized carriers, the current DOP of a spin-VCSEL is limited due to the fast spin flipping and thermal quenching rate in the semiconductor quantum wells (QWs). Hence, integration of spin-decoupled metasurface can be used not only to facilitate chiral emissions with high DOP by filtering out the unwanted CP light, but also to encode both two spin components independently. On the other hand, combing the design of spin-decoupled metasurface with the helicity switching mechanism of spin-VCSELs will give rise to a new type of tunable laser with spin controlled reconfigurable beam functionalities. In addition to shaping the emitted spin photons, the electron spins of the VCSEL can also be controlled using the metasurface in the sense that spatial distributions of the optically injected spin carriers can be manipulated by independently guiding the CP pumping beams of different chirality into specifically assigned regions of the active medium. In this way, spatially selective generation of spin electrons can be implemented inside different areas of semiconductor QWs.”

6. H. Hübener, U. De Giovannini, C. Schäfer, J. Andberger, M. Ruggenthaler, J. Faist, and A. Rubio, "Engineering quantum materials with chiral optical cavities," *Nature materials* **20**, 438-442 (2021).

7. C. Wu, N. Arju, G. Kelp, J. A. Fan, J. Dominguez, E. Gonzales, E. Tutuc, I. Brener, and G. Shvets, "Spectrally selective chiral silicon metasurfaces based on infrared Fano resonances," *Nature communications* **5**, 1-9 (2014).

8. X. Zhang, Y. Liu, J. Han, Y. Kivshar, and Q. Song, "Chiral emission from resonant metasurfaces," *Science* **377**, 1215-1218 (2022).

9. N. Gerhardt, S. Hövel, M. Hofmann, J. Yang, D. Reuter, and A. Wieck, "Enhancement of spin information with vertical cavity surface emitting lasers," *Electronics Letters* **42**, 1 (2006).

10. A. Maksimov, E. Filatov, I. Tartakovskii, V. Kulakovskii, S. Tikhodeev, C. Schneider, and S. Höfling, "Circularly Polarized Laser Emission from an Electrically Pumped Chiral Microcavity," *Physical Review*

Applied **17**, L021001 (2022).

11. J. Frougier, G. Baili, M. Alouini, I. Sagnes, H. Jaffrès, A. Garnache, C. Deranlot, D. Dolfi, and J.-M. George, "Control of light polarization using optically spin-injected vertical external cavity surface emitting lasers," *Applied Physics Letters* **103**, 252402 (2013).

12. J. Sinova, and I. Žutić, "New moves of the spintronics tango," *Nature materials* **11**, 368-371 (2012).

13. N. Nishizawa, K. Nishibayashi, and H. Munekata, "Pure circular polarization electroluminescence at room temperature with spin-polarized light-emitting diodes," *Proceedings of the National Academy of Sciences* **114**, 1783-1788 (2017).

Reviewers' Comments:

Reviewer #1:

Remarks to the Author:

With these revisions and corrections toward my comments/queries as well as other reviewers', the authors have addressed all the comments and points, thus improving the quality of the paper. I am happy to recommend its publication in Nature Communications.

Reviewer #2:

Remarks to the Author:

The authors have addressed most of the questions raised in the first round of review, providing in-depth discussion and convincing new results. In particular, the authors have demonstrated a new device with greatly reduced unwanted zeroth order, which was one of the major issues with the original result. With all the revision and new results, I recommend the publication of the manuscript in Nature Communications.

REVIEWERS' COMMENTS

Reviewer #1:

With these revisions and corrections toward my comments/queries as well as other reviewers', the authors have addressed all the comments and points, thus improving the quality of the paper. I am happy to recommend its publication in Nature Communications.

Response:

We once again thank the reviewer for the constructive suggestions, which help to significantly improve the quality of our work. We are grateful to the reviewer for the positive comment and recommendation on the publication of our revised manuscript.

Reviewer #2:

The authors have addressed most of the questions raised in the first round of review, providing in-depth discussion and convincing new results. In particular, the authors have demonstrated a new device with greatly reduced unwanted zeroth order, which was one of the major issue with the original result. With all the revision and new results, I recommend the publication of the manuscript in Nature Communication

Response:

We are thankful to the reviewer for acknowledging our successful efforts to address the reviewers' questions. We thank the reviewer again for the positive comments and useful suggestions, which allow us to significantly improve the quality of our work. We thank the reviewer for recommending the publication of our revised manuscript.